# Abdominal and Pelvic Floor Activity Related to Respiratory Diaphragmatic Activity in Subjects with and without Non-Specific Low Back Pain

**DOI:** 10.3390/diagnostics12102530

**Published:** 2022-10-18

**Authors:** Cristina Sicilia-Gomez, Samuel Fernández-Carnero, Alicia Martin-Perez, Nicolas Cuenca-Zaldívar, Fermin Naranjo-Cinto, Daniel Pecos-Martín, Maria Cervera-Cano, Susana Nunez-Nagy

**Affiliations:** 1Universidad de Alcalá, Facultad de Medicina y Ciencias de la Salud, Departamento de Enfermería y Fisioterapia, Grupo de Investigación en Fisioterapia y Dolor, 28801 Alcalá de Henares, Spain; 2Research Group in Nursing and Health Care, Puerta de Hierro Health Research Institute-Segovia de Arana (IDIPHISA), 28222 Madrid, Spain

**Keywords:** rehabilitative ultrasound imaging (RUSI), non-specific low back pain (LBP), abdominal wall, pelvic floor (PF), diaphragmatic thickness, diaphragmatic excursion

## Abstract

One of the advances in physiotherapy in recent years is the exploration and treatment by ultrasound imaging. This technique makes it possible to study the relationship between the musculature of the anterolateral wall of the abdomino-pelvic cavity, the pelvic floor muscles and the diaphragm muscle, among others, and thus understand their implication in non-specific low back pain (LBP) in pathological subjects regarding healthy subjects. Objective: To evaluate by RUSI (rehabilitative ultrasound imaging) the muscular thickness at rest of the abdominal wall, the excursion of the pelvic floor and the respiratory diaphragm, as well as to study their activity. Methodology: Two groups of 46 subjects each were established. The variables studied were: non-specific low back pain, thickness and excursion after tidal and forced breathing, pelvic floor (PF) excursion in a contraction and thickness of the external oblique (EO), internal oblique (IO) and transverse (TA) at rest. Design: Cross-sectional observational study. Results: Good-to-excellent reliability for measurements of diaphragm thickness at both tidal volume (TV) (inspiration: 0.763, expiration: 0.788) and expiration at forced volume (FV) (0.763), and good reliability for inspiration at FV (0.631). A correlation was found between the EO muscle and PF musculature with respect to diaphragmatic thickness at TV, inspiration and expiration, and inspiration at FV, in addition to finding significant differences in all these variables in subjects with LBP. Conclusion: Subjects with LBP have less thickness at rest in the OE muscle, less excursion of the pelvic diaphragm, less diaphragmatic thickness at TV, in inspiration and expiration, and in inspiration to FV.

## 1. Introduction

RUSI (rehabilitative ultrasound imaging) is a diagnostic and therapeutic tool which performs its function by imaging tissues, organs and other structures located inside the body. The RUSI technique aims to evaluate the function and morphology of soft tissues and muscles during physical activity or certain tasks [1,2].

In physiotherapy, the most used ultrasound modes are: B-Mode, which shows a large visual field where the anatomical structures are appreciated by means of a gray scale that varies according to the density and location of these structures; and M-Mode, used to obtain the movement patterns of the anatomical structure to be studied, being able to measure the changes that occur in muscle thickness during muscle contraction and relaxation [3].

The studies accessible in the main databases explain the relationship of the musculature of the wall of the abdomen, the pelvic floor and respiratory diaphragm and their implication in non-specific low back pain. Low back pain is one of the main causes of absenteeism from work, and its prevalence throughout life in inhabitants of industrialized countries is over 70% [4]. More than 85% of the patients who come to primary care referring to low back pain do not present a reliable specific cause for their abnormality.

Diaphragmatic movement decreases body pressure by modulating the response of the baroreceptors, located in the circulatory system pathways, on which the negative effect of psychiatric disorders falls, so that the perception of pain is reduced thanks to correct diaphragmatic functioning and excursion [5].

The muscles of the abdominal wall are involved in postural changes. This muscles also develop an antigravitational function; they enlarge their tonic activity as there is an increase in hydraulic pressure exerted by the abdominal contents on the abdominal wall, a consequence of the effect of the gravitational force on the viscera.

This tonic activity helps the rib cage to expand during inspiration and minimizes the final expiratory volume by reducing the volume of the abdominal compartment [6].

The musculature of the abdominal wall and of the pelvic floor (PF) [7] maintains a close relationship with the lumbar, and pubic regions. This musculature, together with the diaphragm [5], has a determining role of mechanical coordination [8] to ensure the stability of the TV and the abdomino-lumbo-pelvic segment. For the action of this musculature of the cavity to be executed in a physiological and adequate manner, it is essential that there be harmony in the totality of the curves of the spine.

In previous studies, it has been concluded by RUSI that there are differences in diaphragmatic thickness in subjects with low back pain with respect to healthy subjects, with subjects with LBP having less diaphragmatic thickness in inspiratory time [9]. Similar changes are even evidenced by MRI, indicating lower excursion and diaphragmatic thickness in subjects with LBP, so it has been previously shown that there are differences in the main respiratory parameters [10].

However, there are no studies where this relationship is demonstrated by the RUSI technique. In addition to all these issues, there is a lack of knowledge of the behavior of the respiratory diaphragm in relation to the musculature of the abdominal wall and the pelvic diaphragm by ultrasound imaging, due to a lack of research. On the other hand, not only is the relationship between these structures unknown in healthy people, but it is also unknown in patients with non-specific low back pain. This is the aim of this investigation.

## 2. Materials and Methods

The selected subjects attended the Faculty of Physiotherapy and Nursing of the University of on a single day to be evaluated.

The inclusion criteria for the group without LBP were: not to present LBP in the last 3 months or at present; aged between 18 and 60 years. The inclusion criteria for the LBP group were: LBP of more than 3 months of evolution; between 18 and 60 years of age; the pain should be localized within the described limits of the lumbar region with or without irradiation to the abdomen, pelvic and/or pubic area. Additionally, exclusion criteria for the two groups were: lumbar or abdominal surgeries; potentially serious medical conditions (neurological, systemic, psychiatric diseases, obesity, intervertebral discs pathologies at any level of the spine or other pathologies such as spondylarthritis or spondylarthrosis, or having experienced a cesarean delivery).

This study had the approval of the Ethics Committee from the University of Alcalá with number CEIM/HU/2019/50. All subjects had to sign an informed consent form and read the Patient Information Sheet to participate in the proposed study. The study was conducted in accordance with the “Guidelines for Reporting Reliability and Agreement Studies” (GRRAS) [11]. In addition, all subjects participating in the study were treated according to the ethical principles established in the World Medical Association (WMA) Declaration of Helsinki.

To calculate the sample size, the formula proposed by Zou et al. (2012) [12] was used with the average data of the 3 interobserver ICC measurements for the variable Diaphragm_VC_inspiration of the first 20 subjects recruited, estimating a final sample of 52 subjects accepting an error α = 0.05 and a power of 80%.

### 2.1. Demographic Variables

The height, weight, body mass index, age, gender, sport practiced and frequency of weekly practice of each patient were collected.

### 2.2. Non-Specific Low Back Pain

This variable was collected through the Visual Numeric Scale, in which the subject evaluates the symptomatology of his pain on a scale of 0 to 10, where 0 is absence of pain, and 10 is the greatest intensity he could feel.

### 2.3. Ultrasound Measurements

In each subject, the muscle thickness of the EO, IO, TrA and diaphragm muscles was measured by ultrasound. For the present study, the diaphragmatic thickening fraction (DTF) will be considered, which corresponds to the difference between the thickness at the end of inspiration and at the end of expiration, compared to the thickness at the end of expiration [13]. A VINNOE35 ultrasound device (Vinno Technology (Suzhou) Co., Ltd., Jiangsu, China) was used for all the sampling with the detailed probes in each region.

Diaphragmatic excursion will be assessed in M-mode with ultrasound, as described under “Ultrasound measurement of the diaphragm”. In addition, PF contraction will be assessed according to the excursion of the musculature from a state of muscle relaxation to maximal voluntary contraction, again using M-mode with ultrasound.

### 2.4. Ultrasound Measurement of the Diaphragm

Subjects will lie supine on the stretcher according to the protocol described by Sarwal A et al. [14]. Transcostal B-mode approach to measure the thickness of the diaphragm (Figure 1) [15], for which the linear transducer with a frequency of 7 to 18 MHz will be placed in the anterior axillary line between the 7th and 8th intercostal space, located in the apposition zone of the diaphragm. The apposition zone is the area of the chest wall where the muscle fibers of the costal portion of the diaphragm run in a cranial direction attached along the inner surface of the rib cage.

The evaluation of the excursion of the diaphragm (Figure 2), an anterior subcostal projection in M-mode will be used with a curvilinear transducer (2 to 6 MHz) placed between the anterior axillary and mid-clavicular lines [16]. Prior to the measurements, the subjects will be asked to breathe at TV for a few minutes. After this, measurements will be taken three times by each of the two investigators, both in TV and in FV, and the mean of these will be obtained. The running time of the ultrasound evaluation will also be estimated [14]. In this study, the right diaphragmatic dome will be evaluated exclusively. It will be performed through the liver window [17].

### 2.5. Ultrasound Measurement of the Abdominal Wall

The patient will be positioned in supine position on the stretcher, in a relaxed position, with the hips flexed 45° and knees resting on a pillow [18]. A linear transducer with frequencies in the range 5–10 MHz will be used for greater accuracy [19]. To assess the lateral musculature (EO, IO, TA), the transducer will be oriented transversally to the longitudinal axis of the patient, in the medial abdominal region, between the costal cartilage of the eleventh rib and the iliac crest (Figure 3) [19].

As the thickness of the lateral abdominal musculature varies according to the phase of the respiratory cycle in which it is found, the measurements will be carried out at the end of a relaxed exhalation with open glottis [19].

For the rectus abdominis (RA), the transducer should be positioned transversely immediately above the navel and slightly offset from the midline (Figure 4) to the right of the subject, until the muscle belly is located on the ultrasound screen.

### 2.6. Ultrasound Measurement of the Pelvic Floor

To assess the function and contraction of the PF musculature, transabdominal ultrasound will be used with the subject placed in the supine position, following previous published protocol [20] (Figure 5). The measurement will be performed with the M-Mode, since it allows observing the contraction of the PF musculature from the initial moment in which the deformation of the same begins to occur in its contractile activity [21]. The limit located between the hypoechoic region that represents the full bladder and the hyperechoic region that corresponds to the PF musculature will be taken as a reference. The purpose is to measure the displacement of the base of the bladder during a voluntary contraction of the PF musculature [20] with previous 500 to 750 mL of water drunk for adequate vision.

### 2.7. Statistical Analysis

Statistical analysis was performed with the program R Ver. 3.5.1. (R Foundation for Statistical Computing, Institute for Statistics and Mathematics, Welthandelsplatz 1, 1020 Vienna, Austria). The significance level was set at *p* < 0.05. The distribution of the variables in each group was tested with the Shapiro–Wilk test which showed many variables with a non-normal distribution which, together with the low sample size (*n* < 30 in each group) made it advisable to use an exact permutation test [22].

Effect size was defined with the nonparametric r statistic as <0.4 (small), 0.4–0.6 (moderate) and >0.6 (large).

The intraclass correlation coefficient (ICC) calculated both intraobserver and interobserver reliability, with the three measurements (m1, m2 and m3) and the two observers, on the variables VC and VF Diaphragm in inspiration and expiration, defined as poor (<0.5), moderate (0.5–0.75), good (0.75–0.9) and excellent (>0.9).

## 3. Results

In this study, 54 subjects were finally analyzed, 23 belonged to the group that presented non-specific low back pain; and 31 subjects belonged to the no pain group. The ‘Table 1’ shows the main characteristics of the subjects, as well as the sport they practice, if so, the frequency with which they practice it, and the level of pain they present, represented through the VNS (Visual Numerical Scale), giving a value from 1 to 10 to their pain, according to its intensity.

The exact permutation test showed significant differences between both groups as shown below (Table 2).

Intraobserver reliability was determined to be good except for the variable Diaphragm VF inspiration which was moderate, as shown in Table 3.

Interobserver reliability was determined to be poor except for the variable Diaphragm VF expiration, which was moderate, as shown in Table 4.

## 4. Discussion

The hypothesis of this study proposes a correlation between the respiratory diaphragm and the musculature of the anterolateral abdominal wall and the PF, which would be altered in those subjects presenting LBP.

The results of this study demonstrate that there are significant differences in the excursion of the pelvic diaphragm, in the thickness of the right external oblique, in the thickness of the respiratory diaphragm in inspiration and expiration in tidal volume, and in inspiration in forced volume, in subjects who present LBP with respect to healthy subjects and, therefore, the DTF of the diaphragm is altered.

It is possible that we did not find significant differences in the variables of diaphragmatic excursion, diaphragmatic thickness in expiration at FV and thickness of the IO, TA and RA in our study due to the limitations encountered in carrying it out, such as the inexperience of the evaluator in the RUSI ultrasound technique, the use of a mask even to perform forced respirations and a small sample size that was insufficient to find any difference in subjects with LBP.

Nowadays, there are studies that demonstrate the existence of a functional deficit in the control of TV by the musculature of the anterolateral wall of the abdomen and the PF, especially the TA muscle, in people suffering from lumbo-pelvic pain [23].

It is worth mentioning a study published in the “British Journal of Sports Medicine” where it was shown that patients with chronic low back pain who performed motor control exercises involving the abdominal wall musculature, saw their pain and disability reduced, in addition to improving their recruitment of the TA [24]. This data agrees with the results obtained in our study, since a significant difference was found in subjects with LBP in one of the abdominal wall muscles, the EO, which raises the idea of a deficient abdomen in this type of subjects and would reinforce the hypothesis of the aforementioned study.

On the other hand, in March 2011, several authors carried out a study with the aim of establishing the ultrasound evaluation of TA thickness during abdominal emptying as a tool to differentiate subjects with low back pain from healthy ones. Although significant differences were observed in the TA contraction index, the relevance was so uncertain that they were barely able to distinguish some subjects from others [25]. This fact supports the results obtained in our study in the permutation test, both in the Monte Carlo Simulation and in the Exact Test, since they reflect a significant difference in the excursion of the pelvic diaphragm and in the thickness of the EO during rest, but not in the TA.

Regarding the involvement of the PF musculature in LBP, in April 2019 [26], an article was published in which an improvement in the function of the PF musculature was evidenced by measuring the displacement of the bladder base by ultrasound imaging (pelvic diaphragm excursion), when subjects were subjected to a program of specific lumbo-pelvic stabilization exercises, which again reinforces the results obtained in this study. 

In 2016, another case–control study was published comparing the activity of the PF musculature when requesting a contraction exclusively of this musculature as a whole in subjects presenting with chronic low back pain with respect to healthy subjects, in which it was concluded that there were statistically non-significant differences in the subjects of the case group, as they showed less displacement of the bladder base in the pelvic diaphragm excursion, compared to controls [27].

Although the results were not statistically significant, perhaps due to the small sample size of the study, this data agrees again with those found in our research, since a difference was also found in the excursion of the pelvic diaphragm in subjects with DLI, so it would be wise in the future to analyze this variable in a study with a larger sample size where we could find relevance with more certainty.

With respect to the respiratory diaphragm, there is a study that analyzed diaphragmatic thickness and excursion, in maximum relaxed inspiration and expiration, in subjects with and without lumbo-pelvic pain, where the result was a decrease in thickness in inspiratory time, when comparing the subjects with each other, which coincides with our study, but exclusively in this variable. Differences were also found in the thickness of the right hemidiaphragm with respect to the left hemidiaphragm in subjects with lumbo-pelvic pain. However, they did not find significant differences in diaphragmatic excursion [9].

It is likely that the results of our study demonstrate significant differences in more variables (TV and FV thickness in expiration and FV thickness in inspiration) since the breaths that the subjects were asked to take are different: maximal relaxed inspiration and expiration in the study mentioned at the beginning of the paragraph; and in our measurements, inspiration and expiration with TV breathing.

Nevertheless, diaphragmatic thickness was measured in both studies in the apposition zone of the diagram, whereas diaphragmatic excursion was performed in the previous study [9,28], which can be compared with our results.

Some studies suggest that the diaphragm muscle is a deep postural stabilizer of the core, associating active breathing with an increase in diaphragmatic thickness and DTF, which leads to an improvement in static balance. A limitation of the diaphragm movement, both in quiet and deep breathing, is related to a balance disorder, thus emphasizing its stabilizing and postural function [28] impact of the diaphragm function. That is why a sudden respiratory movement could lead to a postural change resulting in pain in a subject with LBP, and a limitation of the diaphragm could also lead to a postural disorder causing, again, pain in the subject. Both cases could also imply a limitation in our study, due to not being able to perform forced respirations and even TV adequately.

Another case–control study reflects lower diaphragmatic excursions while breathing at TV in relation to an abnormal posture of this muscle when performing postural tasks and, therefore, an inadequate stabilization of TV by the respiratory diaphragm, in subjects with chronic low back pain, which they propose as the etiology of certain TV disorders [29]. Likewise, some investigations have found less diaphragmatic mobility and resistance in the main respiratory muscles in subjects with LBP [30], which suggests increasingly reduced forced volumes, when requesting several breaths at FV, as was performed in the measurements of our study, perhaps implying less diaphragmatic excursion and less difference between the diaphragmatic thickness in inspiration and expiration at FV.

These theories do not coincide with the results found in our study, since we did not find significant differences in diaphragmatic excursion but did find significant differences in thickness at VF in inspiration. This fact raises the idea of continuing research on diaphragm behavior, where subjects do not have to wear a mask to eliminate possible biases in the measurements, especially in FV.

Regarding the reliability found in other studies, the reference articles show an excellent intraobserver reliability for the measurement of diaphragm thickness, with an ICC between 0.94 in inspiratory time, and 0.98 when performed in expiratory time [15]. In this study, only intraobserver reliability was calculated for diaphragm thickness measurements and was determined to be good to excellent, since the ICC is within 0.8, except for the variable Diaphragm FV inspiration, which would be considered good.

Some of the limitations were: low number of subjects with LBP without spinal pathologies, difficulty in performing VF respirations due to the mandatory use of the mask because of COVID-19, and lack of skill in the motor control of the pelvic diaphragm, which in some cases could prevented them from initiating the activation of the PF musculature.

## 5. Conclusions

Based on the results, it seems that subjects with LBP present: lower thickness in the EO muscle at rest; lower pelvic diaphragm excursion when performing a voluntary contraction; lower thoracic diaphragm muscle thickness during inspiration in FV; lower DTF in TV and FV; and they do not present lower thickness of the abdominal muscles IO, TA and RA, compared to healthy subjects.

In addition, there is a correlation between pelvic diaphragm excursion and thoracic diaphragm thickness in both inspiration and expiration in TV, and during inspiration in FV; there is also a correlation between resting EO thickness and thoracic diaphragm thickness in both inspiration and expiration in TV, and on inspiration in VF.

## Figures and Tables

**Figure 1 diagnostics-12-02530-f001:**
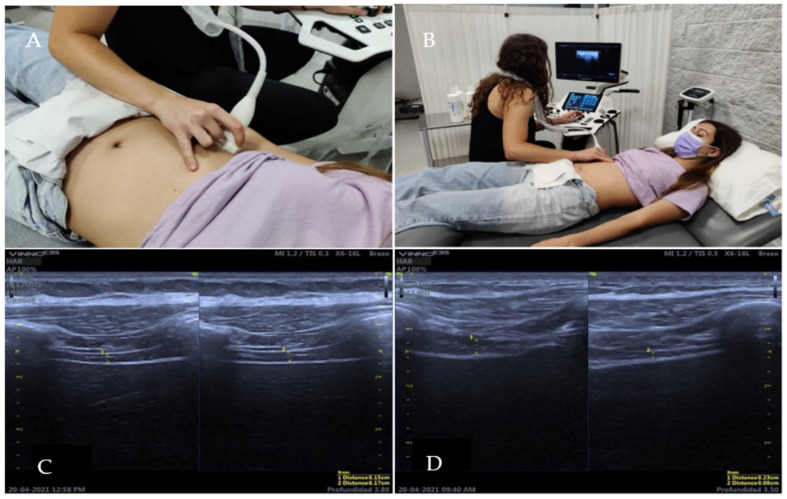
Ultrasound image of diaphragmatic thickness measurement in TV (**A**) and FV (**B**). (**A**) demonstrates transducer placement for an intercostal view, with the transducer positioned on the 9th intercostal space in the anterior axillary line, apposition zone. (**B**) represent the subject and examiner position. (**C**) corresponds to the diaphragmatic thickness when breathing in TV. (**D**) corresponds to the diaphragmatic thickness when breathing in FV.

**Figure 2 diagnostics-12-02530-f002:**
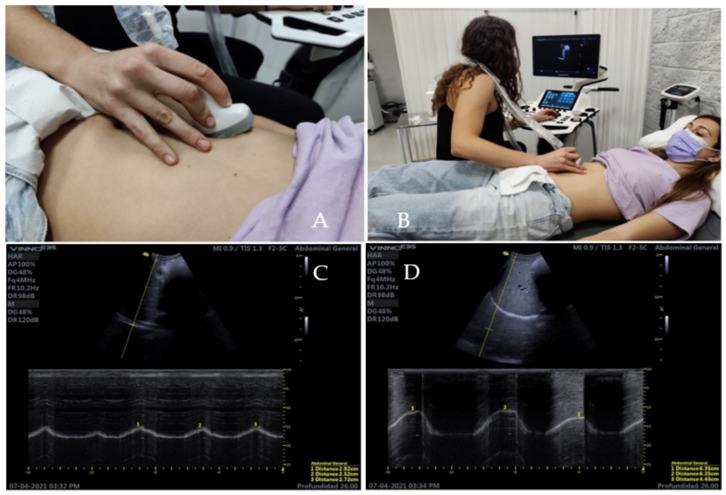
Ultrasound image of diaphragmatic excursion measurement in TV (**A**) and FV (**B**). (**A**) demonstrates transducer placement for an anterior subcostal projection placed between the anterior axillary and mid-clavicular lines. (**B**) represent the subject and examiner position. (**C**) corresponds to the diaphragmatic excursion when breathing in TV. (**D**) corresponds to the diaphragmatic excursion when breathing in FV.

**Figure 3 diagnostics-12-02530-f003:**
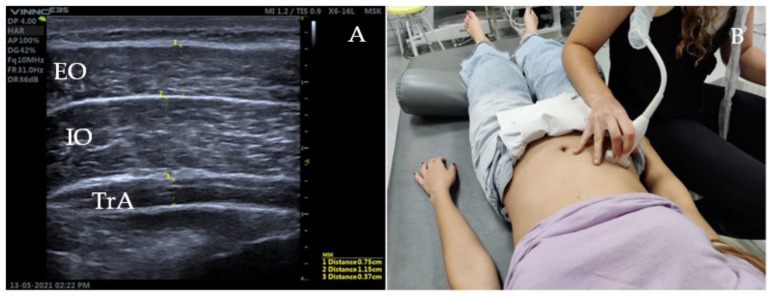
Ultrasound image and ultrasonography of the lateral musculature of the abdominal wall. (**A**) corresponds with the thickness of the lateral abdominal muscles at rest. (**B**) demonstrates the lineal transducer position between the costal cartilage of the eleventh rib and the iliac crest.

**Figure 4 diagnostics-12-02530-f004:**
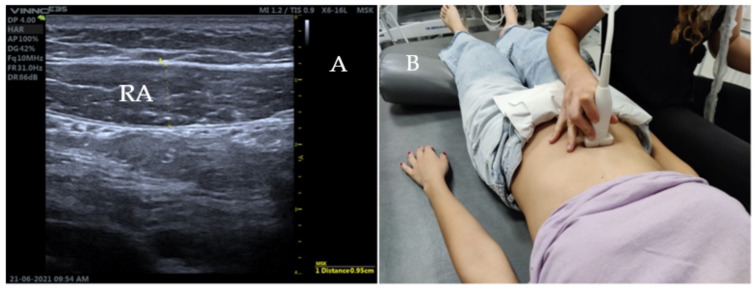
Ultrasound imaging and ultrasonography of the RA. (**A**) represents the RA muscle thickness at rest. (**B**) corresponds to the transducer position above the navel and right by the linea alba.

**Figure 5 diagnostics-12-02530-f005:**
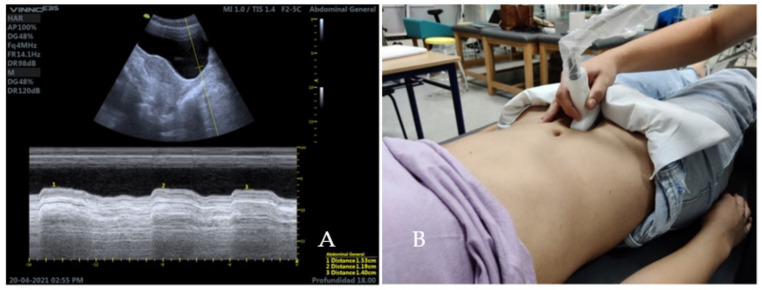
Ultrasound imaging and transabdominal ultrasound of the PF musculature (sagittal plane). (**A**) corresponds to the PF excursion. (**B**) represent the transducer position in a sagittal transabdominal ultrasound imaging of the bladder.

**Table 1 diagnostics-12-02530-t001:** Baseline characteristics of participants.

		With Pain	Without Pain
*n*		23	31
Age		21 [21, 22.5]	21 [21, 22]
Gender, *n* (%)	Men	10 (43.5)	12 (38.7)
	Women	13 (56.5)	19 (61.3)
Height (meters)		1.67 [1.65, 1.75]	1.67 [1.65, 1.78]
Weight (kg)		62 [55, 72]	61 [57, 75]
BMI (Body Mass Index)		21.48 [20.16, 22.94]	22.04 [20.75, 23.7]
Sport, *n* (%)	Dance and fitness	0 (0.0)	1 (3.2)
	Cardio	1 (4.3)	0 (0.0)
	Mountain Cycling	1 (4.3)	0 (0.0)
	Running	1 (4.3)	0 (0.0)
	Crossfit	1 (4.3)	1 (3.2)
	Strength and cardio	0 (0.0)	1 (3.2)
	Football	1 (4.3)	1 (3.2)
	Gym	5 (21.7)	9 (29.0)
	Gym and athletics	0 (0.0)	1 (3.2)
	Gym and climbing	1 (4.3)	0 (0.0)
	Gym and spinning	0 (0.0)	1 (3.2)
	Does not play sports	8 (34.8)	3 (9.7)
	Padel	0 (0.0)	1 (3.2)
	Rugby	0 (0.0)	1 (3.2)
	Running and crossfit	0 (0.0)	1 (3.2)
	Running and bodybuilding	0 (0.0)	2 (6.5)
	Route	1 (4.3)	0 (0.0)
	Softball	1 (4.3)	0 (0.0)
	Triathlon	1 (4.3)	8 (25.8)
	Yoga and running	1 (4.3)	0 (0.0)
Frequency, *n* (%)	1x per week	0 (0.0)	1 (3.2)
	2x per week	2 (8.7)	1 (3.2)
	3x per week	7 (30.4)	5 (16.1)
	4x per week	4 (17.4)	7 (22.6)
	5x per week	0 (0.0)	5 (16.1)
	6x per week	1 (4.3)	7 (22.6)
	7x per week	1 (4.3)	1 (3.2)
	No data	8 (34.8)	4 (12.9)
Non-specific Low Back Pain, *n* (%)	No	0 (0.0)	31 (100.0)
	Yes	23 (100.0)	0 (0.0)
VNS		5.57 ± 1.53	No data

Data expressed as median [interquartile range] or as absolute and relative values (%).

**Table 2 diagnostics-12-02530-t002:** Exact permutation test.

	With Pain	Without Pain	*p* ^a^	Difference (95%CI)	r (95% CI)
	23	31			
Diaphragmatic thickness TV: inspiration	0.17 [0.13, 0.20]	0.14 [0.12, 0.16]	<0.001	0.02 (0, 0.05)	0.26 (0.026, 0.477)
Diaphragmatic thickness TV: respiration	0.14 [0.10, 0.16]	0.12 [0.10, 0.14]	<0.001	0.02 (0, 0.04)	0.242 (0.019, 0.535)
Diaphragmatic thickness FV: inspiration	0.28 [0.23, 0.35]	0.26 [0.20, 0.31]	<0.001	0.04 (−0.01, 0.09)	0.207 (0.022, 0.454)
Diaphragmatic thickness FV: respiration	0.13 [0.11, 0.16]	0.14 [0.11, 0.16]	1	0 (−0.02, 0.02)	0.034 (0.007, 0.279)
DTF TV	0.21 [0.16, 0.30]	0.20 [0.14, 0.25]	<0.001	0.024 (−0.031, 0.081)	0.136 (0.004, 0.374)
DTF FV	1.09 [0.77, 1.51]	0.75 [0.57, 1.33]	<0.001	0.236 (−0.047, 0.5)	0.217 (0.02, 0.449)
Diaphragmatic excursion TV	1.59 [1.38, 2.33]	1.61 [1.29, 2.21]	1	0.08 (−0.27, 0.42)	0.072 (0.008, 0.36)
Diaphragmatic excursion FV	4.37 [3.47, 5.12]	4.38 [3.71, 5.25]	1	−0.092 (−0.75, 0.65)	0.031 (0.002, 0.402)
Pelvic floor excursion	2.83 [2.64, 3.52]	2.97 [2.44, 3.24]	<0.001	−0.08 (−0.25, 0.02)	0.202 (0.013, 0.469)
Transverse abdominis thickness R	2.61 [2.34, 3.12]	2.49 [2.23, 3.02]	1	−0.02 (−0.08, 0.04)	0.095 (0.007, 0.354)
External oblique thickness R	3.01 [2.72, 3.78]	3.04 [2.68, 3.42]	<0.001	0.01 (−0.09, 0.13)	0.019 (0.005, 0.311)
Internal oblique thickness R	2.56 [2.29, 3.23]	2.73 [2.35, 2.99]	1	0.01 (−0.14, 0.16)	0.026 (0.006, 0.264)
Rectus abdominis thickness R	0.17 [0.14, 0.22]	0.18 [0.14, 0.21]	1	0.03 (−0.12, 0.19)	0.056 (0.005, 0.385)

^a^ significant if *p* < 0.05. Data expressed as median [interquartile range] deviation or as absolute and relative values (%). Thickness and excursion variables expressed in centimeters.

**Table 3 diagnostics-12-02530-t003:** ICC results for intraobserver reliability.

	ICC 95%CI
Observer 1: Diaphragm VC inspiration	0.876 (0.813, 0.922)
Observer 1: Diaphragm VC expiration	0.819 (0.732, 0.884)
Observer 1: Diaphragm VF inspiration	0.787 (0.688, 0.863)
Observer 1: Diaphragm VF expiration	0.759 (0.651, 0.844)
Observer 2: Diaphragm VC inspiration	0.763 (0.656, 0.846)
Observer 2: Diaphragm VC expiration	0.788 (0.689, 0.864)
Observer 2: Diaphragm VF inspiration	0.631 (0.489, 0.752)
Observer 2: Diaphragm VF expiration	0.763 (0.657, 0.847)

CI: Confidence Interval.

**Table 4 diagnostics-12-02530-t004:** ICC results for interobserver reliability.

	ICC 95% CI
Diaphragm VC inspiration	0.375 (0.116, 0.585)
Diaphragm VC expiration	0.481 (0.246, 0.663)
Diaphragm VF inspiration	0.245 (−0.024, 0.482)
Diaphragm VF expiration	0.5 (0.267, 0.678)

CI: Confidence Interval.

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
