# Peer review of "Abdominal and Pelvic Floor Activity Related to Respiratory Diaphragmatic Activity in Subjects with and without Non-Specific Low Back Pain"

_diagnostics, 2022, doi:10.3390/diagnostics12102530_

Round 1

Reviewer 1 Report (Previous Reviewer 2)

What is the aim of the study?

The author(s) should write which US machine ( brand, city, country) used for this research?

The discussion is so long please  remove paragraphs starting from lines 300-306 and 312.

Author Response

What is the aim of the study?

Dear reviewer. Thanks for the comment. It is detailed in lines 72-76 and has been remarked in the line 77.

The author(s) should write which US machine ( brand, city, country) used for this research?

Dear reviewer. Thanks for the comment. The US machine, bradn, city… have been detailed in lines 114-115

The discussion is so long please  remove paragraphs starting from lines 300-306 and 312.

Dear reviewer. Thanks for the comment. These paragraphs have been removed.

Reviewer 2 Report (Previous Reviewer 1)

The authors have followed the advice, they have done a good job. The manuscript has been well enriched by aspects of respiratory physiology. 

Author Response

Dear reviewer. Thanks for the comment which encourage for research development.

This manuscript is a resubmission of an earlier submission. The following is a list of the peer review reports and author responses from that submission.

Round 1

Reviewer 1 Report

In this study Dr Sicilia-Gomez-de-Parada and colleagues examine the interest of using rehabilitative ultrasound imaging in the assessment of respiratory muscles thickness and excursion in subjects with non-specific low-back pain. The aim is to show that the relationship of the diaphragm activity and abdominal muscle with the pelvis floor muscle activity is altered in low-back pain patients using rehabilitative ultrasound imaging. The thickness of the different muscle is assessed during quiet and deep breathing. The results show that the thickness of the diaphragm is smaller in subjects without pain during quiet inspiration and expiration and only during forced inspiration. The thickness of the external oblique muscle is also lower. By contrast, pelvis floor excursion is greater in subject without pain.

Some information’s lack in the introduction, what are the roles of the abdominal muscle you choose to record? Are they important for postural adjustment or are they involved in the expiratory phase of breathing?

In introduction it would be interesting to add some functional data on these patients, in particular do they present differences in the main respiratory parameters classically recorded (VT, fR,...). This could lead to a relevant discussion of your results with respect to these data, for example if these patients present a smaller VT this could be related to your data showing that the thickness of the diaphragm or the external oblique is greater. Do deep or forced inhalations or exhalations cause pain in these patients because these exercises mobilize the abdominal muscles and therefore modify the posture ? The respiratory movements induce postural changes, in subjects with low back pain, is the amplitude of postural changes the same, or is it not decreased to limit the movements at the abdominal and pelvic level to avoid the occurrence of pain? It would be interesting to discuss this point.

The discussion could be enriched, you should compare the thickness obtained in your study with the literature data, it’s necessary for the validation of your approach. The thickness may differ depending on which part of the diaphragm is being recorded, so you could expand your discussion with this point.

In the results part, the distribution of the variables are not normal, so it is better to indicate the median with the interquartile range than the mean and SD or SEM, I recommend to make this change in all the tables if the data are in mean and SD or SEM, and to specify in the legends of the tables that the variables are expressed in mean or median.... it is not currently indicated.

Could you specify the breathing task, how many time you recorded quiet breathing or during how many cycles, how many deep breaths were recorded?

Part of the sentence is lost on line 50-51 page 2

Authors should specify which segment (line 57 p2)

In figure 1 and figure 2, C and D are not indicated on the picture

Replace espiration by expiration in table 2

Could you pleased indicate the units of measurement for thickness in table 2 ?

Translate table 3 and table 4 in english

Author Response

Some information’s lack in the introduction,

  1. what are the roles of the abdominal muscle you choose to record?

Dear reviewer. Thanks for the comment. Information has been added from lines 55 to 60.

  1. Are they important for postural adjustment or are they involved in the expiratory phase of breathing?

Dear reviewer. Thanks for the comment. Information has been added from lines 55 to 60.

  1. In introduction it would be interesting to add some functional data on these patients, in particular do they present differences in the main respiratory parameters classically recorded (VT, fR,...).

Dear reviewer. Thanks for the comment. Information has been added from lines 67 to 72.

  1. This could lead to a relevant discussion of your results with respect to these data, for example if these patients present a smaller VT this could be related to your data showing that the thickness of the diaphragm or the external oblique is greater. Do deep or forced inhalations or exhalations cause pain in these patients because these exercises mobilize the abdominal muscles and therefore modify the posture?

Dear reviewer. Thanks for the comment. Information has been added from lines 482-490.

  1. The respiratory movements induce postural changes, in subjects with low back pain, is the amplitude of postural changes the same, or is it not decreased to limit the movements at the abdominal and pelvic level to avoid the occurrence of pain? It would be interesting to discuss this point.

Dear reviewer. Thanks for the comment. Information has been added from lines 482-490.

  1. The discussion could be enriched, you should compare the thickness obtained in your study with the literature data, it’s necessary for the validation of your approach. The thickness may differ depending on which part of the diaphragm is being recorded, so you could expand your discussion with this point.

Dear reviewer. Thanks for the comment. Information has been added from lines 479-481.

  1. In the results part, the distribution of the variables are not normal, so it is better to indicate the median with the interquartile range than the mean and SD or SEM, I recommend to make this change in all the tables if the data are in mean and SD or SEM, and to specify in the legends of the tables that the variables are expressed in mean or median.... it is not currently indicated.

Dear reviewer. Thanks for the comment. The changes in tables have been done.

  1. Could you specify the breathing task, how many time you recorded quiet breathing or during how many cycles, how many deep breaths were recorded?

Dear reviewer. Thanks for the comment. The information requested is detailed in lines 131 to 133.

  1. Part of the sentence is lost on line 50-51 page 2

Dear reviewer. Thanks for the comment. The sentence has been retrieved and indicated where it belongs.

  1. Authors should specify which segment (line 57 p2)

Dear reviewer. Thanks for the comment. It has been specified.

  1. In figure 1 and figure 2, C and D are not indicated on the picture

Dear reviewer. Thanks for the comment. It has been enhanced and indicated.

  1. Replace espiration by expiration in table 2

Dear reviewer. Thanks for the comment. It has been adapted in English.

  1. Could you pleased indicate the units of measurement for thickness in table 2 ?

Dear reviewer. Thanks for the comment. It has been detailed.

  1. Translate table 3 and table 4 in English

Dear reviewer. Thanks for the comment. The tables 3 and 4 has been adapted in English.

Reviewer 2 Report

The sentences of Table 3 and 4 should be written in English. EO and IO muscles should be written open.

Author Response

Dear reviewer. Thanks for the comment. The tables 3 and 4 has been adapted in English.